# DIVERGE: DIVERSITY-ENHANCED RAG FOR OPEN-ENDED INFORMATION SEEKING

## ABSTRACT

Existing retrieval-augmented generation (RAG) systems are primarily designed under the assumption that each query has a single correct answer. This overlooks common information-seeking scenarios with multiple plausible answers, where diversity is essential to avoid collapsing to a single dominant response, thereby constraining creativity and compromising fair and inclusive information access. Our analysis reveals a commonly overlooked limitation of standard RAG systems: they underutilize retrieved context diversity, such that increasing retrieval diversity alone does not yield diverse generations. To address this limitation, we propose **DIVERGE**, a plug-and-play agentic RAG framework with novel reflection-guided generation and memory-augmented iterative refinement, which promotes diverse viewpoints while preserving answer quality. We introduce novel metrics tailored to evaluating the diversity–quality trade-off in open-ended questions, and show that they correlate well with human judgments. We demonstrate that DIVERGE achieves the best diversity–quality trade-off compared to competitive baselines and previous SOTA methods on the real-world `Infinity-Chat` dataset, substantially improving diversity while maintaining quality. More broadly, our results reveal a systematic limitation of current LLM-based systems for open-ended information-seeking and show that explicitly modeling diversity can mitigate it. Our code is available at: `https://github.com/au-clan/diverge`.

## 1 INTRODUCTION

Retrieval-Augmented Generation (RAG) Lewis et al. (2020) enhances LLMs' ability to ground responses in up-to-date external knowledge and improve response quality in knowledge-intensive tasks. However, most prior works Zhang et al. (2025c); Asai et al. (2024); Yang et al. (2018); Yu et al. (2024) are built upon the hypothesis that *each question has a single, clearly defined factual answer*. While this hypothesis enables effective factual grounding, it overlooks the fact that real-world information-seeking needs are often open-ended and admit *multiple plausible answers* Wikimedia Foundation (2018); Arora et al. (2022); Arora (2024); Jiang et al. (2025), as cultural backgrounds Hershcovich et al. (2022), values Solaiman & Dennison (2021), and personal preferences Sorensen et al. (2024) equip individuals with diverse perspectives when seeking information.

In open-ended settings, response diversity is a key evaluation criterion, supporting fair and inclusive representation of diverse viewpoints and mitigating the risk that homogenized LLM outputs narrow human creativity Röttger et al. (2025); Zhang et al. (2025d). At the same time, ensuring high answer quality while promoting diversity remains a key challenge Lanchantin et al. (2025); Shypula et al. (2025). Prior work shows that current close-book LLMs, shaped by post-training objectives, often overlook output diversity, resulting in homogenized generation regimes Wright et al. (2025); Jiang et al. (2025). Although result diversification has been studied in information retrieval (IR) Khan et al. (2013), simply integrating diverse IR techniques into RAG pipelines does not guarantee diverse generations, leaving it unclear whether RAG systems can overcome the inherent homogenization tendencies of current LLMs.

**Challenges.** As a concrete illustration of these challenges, Fig. 1 shows a key limitation of RAG for open-ended questions, where diverse retrieved contexts fail to translate into diverse outputs due to LLM homogenization. Our investigation (§ 6.4) further shows that simple strategies are insufficient: increasing retrieval diversity alone does not lead to more diverse generations, while state-of-the-art

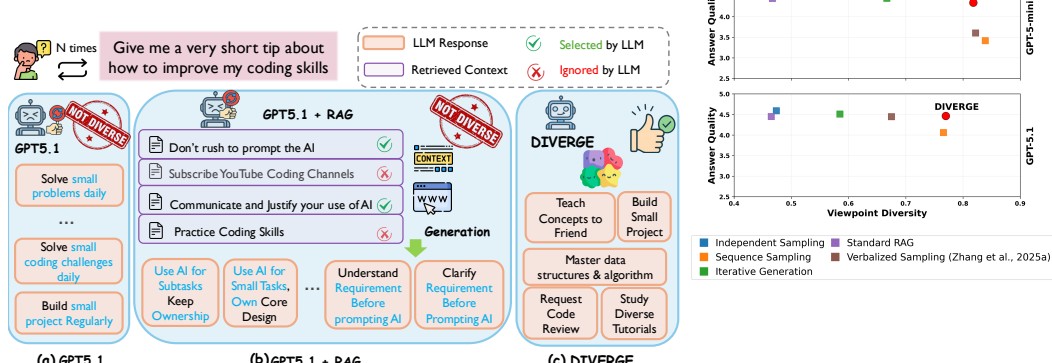

Figure 1: **Left**: *Illustrative example* of an open-ended information-seeking query. (a) LLMs exhibit homogenized (blue) outputs, (b) standard RAG still produces repetitive responses even when the retrieved contexts contain diverse evidence. In contrast, (c) DIVERGE generates diverse outputs while maintaining high answer quality. **Right**: Diversity–quality trade-off of different methods. *Upper-Right* indicates better. DIVERGE achieves the *best performance* among all methods.

prompt-based LLM baselines for diversity-enhancement Zhang et al. (2025b) achieve limited diversity gains at the cost of substantial quality degradation in open-ended information-seeking.

Motivated by these observations, we conduct a systematic analysis of the diversity challenges (detailed in § 3.2) faced by existing RAG systems. In particular, we identify three main issues: (C1) **Single-Answer Bias:** induced by prevailing RAG paradigms, which encourages overconfident generation and causes models to overlook alternative yet plausible answers; (C2) **Missing Diversity Preservation:** reflecting the lack of mechanisms for long-horizon diversity preservation and leading to highly similar outputs across responses; and (C3) **Limited Practical Applicability:** Most existing solutions rely on access to token-level logits and are therefore largely incompatible with frontier LLMs.

**Present work.** To address these challenges, we propose **DIVERGE** (**Div**ersity-**E**nhanced **R**etrieval-Augmented **Ge**neration), the first plug-and-play agentic RAG framework explicitly designed to address the diversity–quality trade-off in real-world open-ended information-seeking (§ 4), equipped with novel components that explicitly promote diversity while preserving answer quality. Specifically, DIVERGE addresses the three challenges by: (i) mitigating the single-answer bias (*C1*) through explicit reflection on uncovered viewpoints; (ii) enabling long-horizon diversity preservation while maintaining answer quality (*C2*) via an iterative RAG process with lightweight memory and evidence-grounded generation; (iii) avoiding reliance on token-level logits (*C3*), thereby ensuring compatibility with arbitrary LLM backbones, including closed-source frontier models.

Most existing RAG evaluation metrics Es et al. (2024) rely on predefined ground-truth answers and therefore do not scale to open-ended settings. Moreover, existing diversity–quality trade-off evaluations largely focus on creative tasks Lanchantin et al. (2025), leaving a gap for information-seeking scenarios. To facilitate this evaluation, we introduce a novel set of metrics. To capture both high-level diversity and the diversity across multiple viewpoints within a single response, we consider two complementary dimensions: *semantic diversity*, which measures diversity at the level of the overall response, and *viewpoint diversity*, which decomposes a response into a set of atomic viewpoints and measures diversity across them. For *quality*, given the open-ended nature of the task and aligned with convention Badshah & Sajjad (2024); Xu et al. (2025), we adopt an LLM-as-a-judge paradigm. Finally, to enable intuitive comparison of trade-off performance across models, we propose a *Unified Diversity–Quality Harmonic Score* (*Unified Score*).

We empirically validate DIVERGE on Infinity-Chat, a complex real-world open-ended benchmark Jiang et al. (2025). DIVERGE achieves the **highest** *Unified Score* across all methods, improving *semantic diversity* by $\sim 2.5\times$ and *viewpoint diversity* by $\sim 1.6\times$ over direct prompting, with only negligible impact on answer quality. We further demonstrate the effectiveness of our framework through ablation studies and provide in-depth analyses (§ 6.5 and § 7).

Our contributions can be summarized as follows:

• We identify a commonly overlooked limitation of standard RAG systems. In open-ended information-seeking settings, they suffer from knowledge collapse and make limited use of diverse

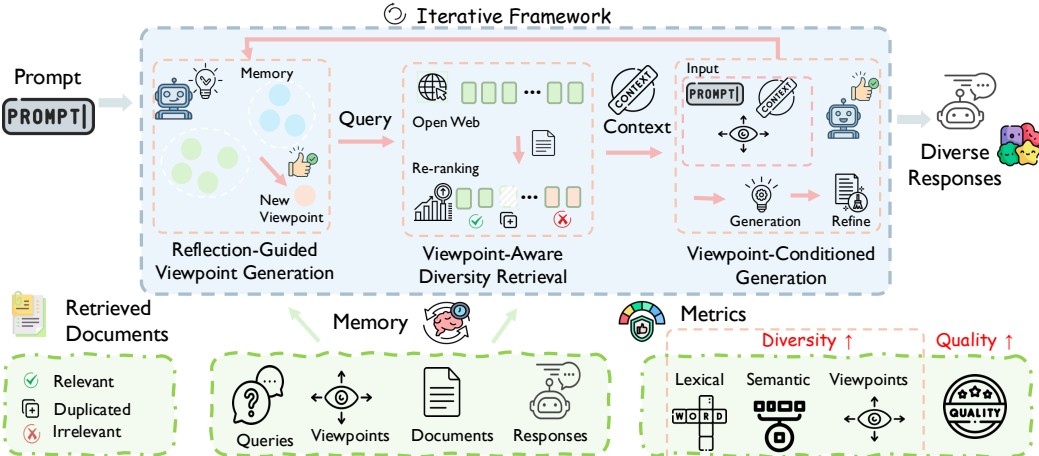

Figure 2: Overview of DIVERGE, a plug-and-play agentic RAG framework for open-ended settings that promotes diverse viewpoints via reflection-guided viewpoint generation and viewpoint-conditioned retrieval and generation, with broad LLM compatibility.

retrieved contexts. On the other hand, existing state-of-the-art prompt-based methods for diversity incur substantial quality degradation, leaving the problem unresolved.

- We propose a set of evaluation metrics for open-ended information-seeking, focusing on the diversity–quality trade-off, enabling intuitive and systematic comparison.

- We introduce DIVERGE, a plug-and-play diversity-enhanced agentic RAG framework supporting frontier closed-source LLMs, and empirically show it achieves the best diversity–quality trade-off in real-world settings.

## 2 RELATED WORK

**Homogeneity of LLMs.** Recent studies Jiang et al. (2025); Zhang et al. (2025d) show that compared to human authors, LLMs generate significantly less diverse outputs. This homogeneity has raised broad concerns, including the risk of social and cultural biases induced by dominant perspectives (Röttger et al., 2025), the potential for epistemic collapse (Wright et al., 2025), failures in customizable AI systems (Zhang et al., 2025d), and the homogenization of human thinking under exposure to LLM-generated content Jiang et al. (2025). Prior work suggests that such homogeneity is largely driven by post-training objectives that encourage models to sharpen their output probability distributions (Lanchantin et al., 2025). In addition, preference data may systematically reward more typical responses, further biasing models toward less diverse outputs Zhang et al. (2025b). RAG has the potential to access more diverse knowledge than parametric models Wright et al. (2025); However, it remains fundamentally constrained by the homogenized generation of its backbone LLMs, which favor deterministic outputs Zharzhavsky et al. (2026) over alternatives. As a result, whether RAG can produce diverse responses in open-ended settings remains largely unexplored.

**Techniques for Increasing Generation Diversity.** A wide range of decoding-time strategies has been proposed to increase the generation diversity of LLM, primarily by adjusting stochastic sampling hyperparameters such as temperature, top-$p$, top-$k$ (Shi et al., 2024), and min-$p$ Nguyen et al. (2024). However, these approaches offer only limited improvements when LLMs exhibit collapsed output distributions Jiang et al. (2025), and many recent close-sourced frontier LLMs no longer support such decoding controls, including models such as GPT-5 and o3. Another line of work seeks to improve diversity by retraining LLMs with diversity-aware alignment objectives, such as *DivPO* Lanchantin et al. (2025). While effective, these methods are resource-intensive, require training from scratch, and cannot be applied to frontier closed-source LLMs. Finally, prompt-based approaches have been explored to elicit more diverse outputs Shur-Ofry et al. (2024); Zhang et al. (2025b). In practice, however, such methods often achieve higher diversity at the expense of answer quality (cf. § 6.5 for details), which is particularly undesirable for information-seeking tasks. Taken together, these limitations motivate the need for alternative approaches that can improve diversity without compromising answer quality, particularly for real-world information-seeking tasks in RAG.

**Diversity in IR and RAG.** In IR, diversity has long been used to cover a broader range of user preferences through techniques such as query rewriting and re-ranking (Mohankumar et al., 2021; Krestel & Fankhauser, 2012). However, traditional IR systems typically return a ranked list of documents, leaving users to manually interpret the retrieved information to satisfy their needs Li et al. (2025). This limitation motivates RAG, which directly integrates retrieved evidence into responses.

In RAG, existing work on diversity primarily focuses on retrieving diverse contexts to support QA tasks, such as mitigating context window limits Wang et al. (2025) or enabling multi-hop reasoning Rezaei & Dieng (2025). Systems such as DeepResearch Xu & Peng (2025) similarly focus on aggregating evidence to match a single predefined answer. All these approaches and their evaluations largely retain the single-answer assumption and aim to improve correctness. In contrast, our work targets open-ended settings with multiple valid answers, explicitly promotes output diversity, and evaluates diversity at the level of final responses. Other related studies either focus on narrow creative domains, such as recipe cross-cultural adaptation Hu et al. (2025), or briefly discuss the role of RAG in the context of knowledge collapse Wright et al. (2025). As a result, the design and evaluation of RAG systems that explicitly target output diversity in open-ended information-seeking settings remains largely unexplored, which is our focus.

## 3 TASK: OPEN-ENDED INFORMATION SEEKING

### 3.1 TASK FORMULATION

Given an arbitrary diversity metric $\mathcal{D}$, quality metric $\mathcal{Q}$, and an arbitrary model configuration $c$, the task takes as input a set of open-ended queries $\mathbb{Q} = \{q^1, q^2, \ldots, q^N\}$. For each query $q^i$, the model produces a set of $K$ responses, we denote this set by $\mathcal{A}_c^i = \{a_{c,1}^i, a_{c,2}^i, \ldots, a_{c,K}^i\}$. The objective of the task is to produce responses that collectively exhibit both high diversity and quality. Quality can be easily assessed by averaging the quality of each output. So the primary question is defining diversity.

Kirk et al. (2023) propose two paradigms for measuring diversity: *across-input*, which considers variation across different input–output pairs, and *per-input* diversity, which captures diversity only among multiple outputs generated for the same input. We adopt the *per-input* paradigm, as our focus is on assessing the diversity of responses generated for the same open-ended query. So defined as:

$$\text{Diversity}_{\mathcal{D}}(c) := \frac{1}{N} \sum_{i=1}^{N} \mathcal{D}(\mathcal{A}_c^i).$$

### 3.2 DIVERSITY CHALLENGES FOR RAG

**C1: Single-Answer Bias.** Existing RAG systems are optimized to produce reliable and accurate answers under a single-answer assumption Asai et al. (2024); Zhang et al. (2025a), which typically leads to low uncertainty across multiple generations Zharzhavsky et al. (2026), even less variation than the already highly homogenized underlying LLMs Soudani et al. (2025). However, this bias limits diversity in open-ended settings: LLMs tend to prioritize a narrow, high-confidence subset of contexts and ignore alternative yet plausible information Hu et al. (2025).

**C2: Missing Diversity Preservation Across Generations.** Existing RAG mechanisms struggle to preserve diversity across multiple generations, often producing highly similar outputs due to the lack of explicit mechanisms for summarizing, compressing, and retaining previously generated information in support of diversity. Moreover, open-ended questions typically involve a lot of viewpoints Jiang et al. (2025) distributed across many sources, making single-shot retrieval insufficient for covering the rich information space.

**C3: Limited Practical Applicability.** Most existing test-time diversity-enhancing approaches Vijayakumar et al. (2016); Nguyen et al. (2024); Shi et al. (2024) rely on decoding strategies. While these methods have shown effectiveness, they typically require access to token-level logits during generation, which remains unavailable in most closed-source frontier LLMs Hiranandani et al. (2025). Furthermore, an emerging trend among frontier models is to prohibit the use of decoding hyperparameters such as temperature OpenAI Community (2025), as observed in recent models such as GPT-5, o3, and subsequent variants, further limiting the applicability of these approaches in real-world.

# 4 DIVERGE

Motivated by *Plan-and-Solve* agentic designs Wang et al. (2023), we introduce a diversity-oriented RAG process that iteratively summarizes, reflects, generates new viewpoints, retrieves evidence, and refines responses, enabling a better balance between diversity and answer quality.

**DIVERGE** (**Div**ersity-**E**nhanced **R**etrieval-Augmented **Ge**neration) is a plug-and-play agentic RAG framework which explicitly models diverse viewpoints to address the single-answer bias (C1). It further mitigates diversity collapse (C2) by introducing an iterative RAG framework with a lightweight memory. Finally, DIVERGE does not rely on access to token logits, allowing it to be applied with *any* LLM as the backbone and enabling strong practical applicability (C3). Figure 2 shows the framework, and Algorithm 1 in the Appendix describes the entire procedure.

**Reflection-Guided Viewpoint Generation.** Prior research Wang et al. (2022) suggests that multiple viable internal reasoning trajectories can coexist within LLMs, and that appropriate prompting can steer models toward different directions Zhuo et al. (2024). Moreover, mechanistic analyses indicate that multiple latent features coexist within models and can be selectively activated Anthropic (2023). Inspired by these insights, we conceptualize these latent features as *viewpoints* and leverage them as a core abstraction in the design of our framework.

DIVERGE first summarizes the initial RAG response and then iteratively reflects on prior outputs to maintain a set of existing viewpoints. At each iteration, the LLM identifies a new, insufficiently covered viewpoint based on those previously explored, thereby avoiding repeated generation at a high level. This reflection-guided process promotes the exploration of alternative perspectives and mitigates the tendency to repeatedly generate responses from a single dominant stance.

**Viewpoint-Aware Diversity Retrieval.** While viewpoints encourage considering a problem from multiple perspectives, they are inherently hypothetical and lack factual grounding, and the underlying model may not possess sufficient knowledge to support all generated viewpoints. To address this, we incorporate a retrieval mechanism that queries the open web for relevant evidence and applies diversity-aware re-ranking, enabling the LLM to generate diverse and factually grounded responses under specific viewpoints.

Conditioned on a given viewpoint, the LLM generates a query that is issued to a web agent to retrieve documents from the open web. The retriever then performs diversity-aware re-ranking by jointly considering relevance to the current query, diversity with respect to previously retrieved contexts, and diversity among candidates selected within the current iteration. This design enables evidence retrieval that supports the current viewpoint while avoiding redundancy with evidence and viewpoints explored in earlier iterations. Specifically, we extend Maximal Marginal Relevance (MMR) Carbonell & Goldstein (1998) with an iteration-aware formulation that accounts for similarity to documents retrieved in previous iterations, defined as:

$$s_t(d) \;=\; \alpha \cdot \mathrm{Rel}(d, q_t) \;-\; \beta \cdot \max_{h \in \mathcal{M}_{<t}} \mathrm{Sim}(d, h) - (1 - \alpha) \cdot \max_{s \in \mathcal{S}_t} \mathrm{Sim}(d, s).$$

Here, $t$ denotes the current iteration, $q_t$ is the viewpoint-conditioned query at iteration $t$. $\mathrm{Rel}(d, q)$ denotes the relevance score between document $d$ and the current query. $\mathcal{M}_{<t}$ denotes the memory containing contexts retrieved in all previous iterations, and $\mathcal{S}_t$ denotes the set of documents already selected within the current iteration, and $\mathrm{Sim}()$ denotes cosine similarity between document embeddings. $\alpha$ and $\beta$ are tunable hyperparameters that control the trade-off between relevance and diversity.

**Viewpoint-Conditioned Generation.** Even when a novel viewpoint is identified and supported by factual evidence, the result may still be unsatisfactory, as it may lack sufficient connection from the user's original query to the targeted viewpoint or omit essential elements required to fully address the user's needs. To bridge this gap, we introduce a viewpoint-conditioned generation and refinement process. Specifically, the generation step is explicitly conditioned on both the original query and the targeted viewpoint, encouraging the model to approach the question from the specified perspective. We further refine the generated response to ensure it remains well aligned with the original query and maintains coherent logical connections, preventing excessive deviation while improving completeness.

Table 1: Evaluation of diversity and quality on the `Infinity-Chat`. (1) For the diversity–quality trade-off, using *Independent* generation as the baseline. Red indicates improvements and blue indicates degradations relative, darker colors indicate larger effect sizes. DIVERGE is the *only* method that improves diversity while maintaining comparable quality. (2) *Unified Score*, highlighted in purple, provides an overall comparison by jointly accounting for diversity and quality, with best shown in **bold**. DIVERGE achieves the strongest performance across all models.

| Methods | GPT-5-mini | | | | | GPT-5.1 | | | | |
| --- | --- | --- | --- | --- | --- | --- | --- | --- | --- | --- |
| | Diversity ↑ | | Quality ↑ | Unified Score ↑ | | Diversity ↑ | | Quality ↑ | Unified Score ↑ | |
| | $\mathcal{D}_{\text{Sem}}$ | $\mathcal{D}_{\text{View}}$ | | $\text{U}_Q^{\text{Sem}}$ | $\text{U}_Q^{\text{View}}$ | $\mathcal{D}_{\text{Sem}}$ | $\mathcal{D}_{\text{View}}$ | | $\text{U}_Q^{\text{Sem}}$ | $\text{U}_Q^{\text{View}}$ |
| **Closed-Book LLMs** | | | | | | | | | | |
| Independent Sampling | 0.100 | 0.510 | 4.578 | 0.119 | 0.417 | 0.096 | 0.474 | 4.590 | 0.094 | 0.346 |
| List Generation | 0.446 | 0.839 | 3.417 | 0.167 | 0.160 | 0.309 | 0.766 | 4.059 | 0.456 | 0.518 |
| Iterative Generation | 0.176 | 0.667 | 4.449 | 0.324 | 0.556 | 0.131 | 0.585 | 4.510 | 0.198 | 0.450 |
| Verbalized Sampling Zhang et al. (2025b) | 0.417 | 0.822 | 3.603 | 0.292 | 0.273 | 0.217 | 0.675 | 4.447 | 0.425 | 0.586 |
| **RAGs** | | | | | | | | | | |
| Vanilla RAG | 0.106 | 0.467 | 4.444 | 0.132 | 0.319 | 0.107 | 0.465 | 4.449 | 0.148 | 0.329 |
|   + Diverse Re-ranking | 0.106 | 0.469 | 4.465 | 0.145 | 0.330 | 0.109 | 0.475 | 4.428 | 0.151 | 0.334 |
|   + Contexts Shuffle | 0.116 | 0.487 | 4.429 | 0.172 | 0.361 | 0.119 | 0.475 | 4.422 | 0.179 | 0.340 |
|   + Multi-Query | 0.100 | 0.446 | 4.423 | 0.124 | 0.283 | 0.102 | 0.445 | 4.452 | 0.137 | 0.292 |
|   + All | 0.110 | 0.466 | 4.464 | 0.159 | 0.322 | 0.110 | 0.471 | 4.532 | 0.169 | 0.353 |
| **DIVERGE** | 0.269 | 0.818 | 4.342 | **0.557** | **0.728** | 0.219 | 0.770 | 4.462 | **0.473** | **0.713** |

Together, these components equip DIVERGE to explore diverse viewpoints while maintaining a high level of output quality. Empirical results are presented in Section 6.5. Details of implementation are shown in Appendix B.

# 5 EVALUATING QUALITY–DIVERSITY TRADE-OFFS

Our evaluation focuses on *diversity* (§ 5.1), *quality* (§ 5.2), and their trade-off (§ 5.3), as diversity is only meaningful when quality is preserved Lanchantin et al. (2025).

## 5.1 DIVERSITY METRICS

**Semantic Diversity.** Semantic diversity Guo et al. (2024) measures variation in meaning among generated responses. For each input, we compute semantic diversity as the average pairwise cosine distance between answer embeddings, yielding a normalized score in $[0, 1]$, where lower similarity corresponds to higher diversity Stasaski & Hearst (2023).

$$\mathcal{D}_{\text{sem}}(\mathcal{A}_c^i) = \frac{1}{\binom{K}{2}} \sum_{1 \le j,k \le K} \frac{1 - d_{\cos}\Big(e(a_{c,j}^i), e(a_{c,k}^i)\Big)}{2},$$

where $e(\cdot)$ denotes the embedding of a generated answer.

**Viewpoint Diversity.** While semantic diversity captures variation across responses, it is limited in identifying multiple implicit viewpoints within a single response. This is particularly problematic for open-ended questions, where one answer may encompass several distinct viewpoints (example is provided in Appendix G), leading semantic metrics to underestimate viewpoint-level diversity.

To address this gap, inspired by prior work Wright et al. (2025) that considers diversity at the level of viewpoints, we adopt a simplified formulation tailored to open-ended generation and propose a *viewpoint diversity* metric. Given that different questions may entail different intrinsic requirements on the number of viewpoints (e.g., requesting multiple suggestions versus a single one), we define viewpoint diversity as the fraction of mutually non-overlapping viewpoints, which normalizes variation across queries with differing viewpoint demands. Specifically, we apply an automatic claim extraction function $f$ via an LLM to decompose each generated response into a set of *atomic claims*. Each atomic claim corresponds to a minimal viewpoint that independently addresses a specific aspect of the query. We then aggregate all extracted claims across the $K$ responses:

$$\mathcal{D}_{\text{view}}(\mathcal{A}_c^i) = \frac{\big|\text{unique}\big(\mathcal{C}_c^i\big)\big|}{k_i}, \mathcal{C}_c^i = \bigcup_{j=1}^{K} f\big(a_{c,j}^i\big) = \{c_{c,1}^i, c_{c,2}^i, \dots, c_{c,k_i}^i\},$$

where $k_i$ denotes the number of claims for query $i$, and $|\text{unique}(\cdot)|$ denotes the number of claims whose pairwise embedding similarity falls below a predefined threshold, indicating distinct viewpoints. Additional details are reported in Appendix C.1.

## 5.2 QUALITY METRIC

**Quality Score.** Given that these open-ended questions admit a large number of reasonable answers, collecting all valid responses is impractical. As a result, metrics that rely on predefined ground-truth answers, such as *Factual Correctness* or *Answer Accuracy* Es et al. (2024), are not applicable in our setting. Following prior work Badshah & Sajjad (2024); Yu et al. (2025); Gu et al. (2024), we adopt an *LLM-as-a-Judge* framework to assess quality, as it has been shown to exhibit stronger judgment capabilities than *reward models* in settings with high uncertainty and without explicit ground-truth answers Xu et al. (2025). We evaluate response quality along four dimensions: *factual accuracy*, *evidence support*, *internal consistency*, and *question relevance*. Judgments are reported on a five-level ordinal scale, with higher scores indicating better quality. Details are described in Appendix C.2.

**Human Annotation Agreement.** We compute the agreement between Quality Scores and the mean human annotations on a pre-collected human-labeled `Infinity-Chat` dataset[1] consisting of 1,500 samples annotated by 25 annotators. Using the quadratic-weighted Cohen's kappa McHugh (2012), our proposed quality metric achieves a score of 0.54, which is slightly lower than the agreement between individual human annotators and the mean human annotations (0.61), but higher than the average inter-annotator agreement under random pairing (0.44). Given the inherently subjective nature of the task, these results suggest that our metric exhibits a *reasonable level* of alignment with human judgments. Details are reported in Appendix C.3.

## 5.3 UNIFIED SCORE

Diversity and quality naturally form a trade-off, which poses challenges for model comparison. Inspired by prior work on harmonic aggregation for balancing competing objectives Sasaki et al. (2007); Min et al. (2020), we introduce the *Unified Diversity–Quality Harmonic Score* metric for capturing the diversity–quality trade-off in open-ended question answering. Specifically, it is defined as the harmonic mean of normalized quality and diversity at the query level:

$$\mathrm{U}_{\mathrm{Q}}^{\mathrm{D}} = \frac{1}{N} \sum_{i=1}^{N} \mathcal{U}_{\mathrm{Q}}^{\mathrm{D}} \left( \mathcal{A}_c^i \right) = \frac{1}{N} \sum_{i=1}^{N} \frac{2 \cdot \tilde{Q} \left( \mathcal{A}_c^i \right) \cdot \tilde{D} \left( \mathcal{A}_c^i \right)}{\tilde{Q} \left( \mathcal{A}_c^i \right) + \tilde{D} \left( \mathcal{A}_c^i \right)},$$

where $\tilde{Q}^i$ and $\tilde{D}^i$ denote the query-wise min-max normalized quality and diversity scores in $[0, 1]$, respectively. Accordingly, we obtain two variants, $\mathrm{U}_{\mathrm{Q}}^{\mathrm{Sem}}$ and $\mathrm{U}_{\mathrm{Q}}^{\mathrm{View}}$, by applying diversity with semantic and viewpoint measures.

## 6 EXPERIMENT

### 6.1 DATASET

We use the `Infinity-Chat` Jiang et al. (2025) dataset[2], which contains large-scale, real-world open-ended user queries with multiple plausible answers and no ground truth. We select queries from three categories: *Alternative Perspectives*, *Ideation and Brainstorming*, and *Information-Seeking*, that reflect diverse user information-seeking needs. These categories span 10 fine-grained user intents (e.g., *Decision Support* and *Controversial Questions*), representing common scenarios where multiple viewpoints are expected. In contrast, we do not use datasets such as NoveltyBench Zhang et al. (2025d) or CoverageQA Wong et al. (2024), as the open-ended questions in these manually curated benchmarks are relatively simple and can often be answered without requiring additional retrieval. As a result, they are less representative of realistic RAG scenarios that need external knowledge. Additional details are provided in Appendix D.1.

### 6.2 BASELINES

Our baselines fall into two categories: (1) *Closed-Book LLMs*, which include LLMs without retrieval and prompt-based strategies designed to encourage diversity Zhang et al. (2025b); and (2) *Baselines*

---

[1]Dataset URL: LINK (cf. § 6.1 for details).
[2]Link

*with retrieval*, including vanilla RAG and variants that incorporate simple diversity-enhancing strategies. Examples with detailed descriptions are shown in Appendix D.2.

**Closed-Book LLMs.** We consider Independent Sampling generation as the basic baseline, where the LLM is run independently $K$ times with one response each time. We also explore three prompt-based strategies for increasing diversity: (1) List Generation runs the LLM once and explicitly prompts it to return $K$ distinct responses. (2) Iterative Generation simulates an interactive dialogue by iteratively appending previous responses to the history and prompting the model to generate new answers. (3) Verbalized Sampling Zhang et al. (2025b) is a state-of-the-art strategy that prompts the model to produce $K$ distinct candidates along with their verbalized probabilities in a single generation, thereby mitigating collapse toward typical outputs and encouraging greater diversity.

**Baselines with retrieval.** Since existing RAG methods are not designed for open-ended generation with diverse outputs, we construct strong retrieval-based baselines by adapting widely used diversity-enhancing strategies from IR. Specifically, we consider Vanilla RAG as a standard baseline, together with four strategies: (1) Diversity Reranking, which applies classical MMR Carbonell & Goldstein (1998) to promote ranking diversity; (2) Context Shuffle, which mitigates positional bias Liu et al. (2023) by randomly shuffling contexts at each generation; (3) Multi-Query, which performs retrieval using multiple LLM-generated query rewrites; and (4) All, which combines all the above strategies.

## 6.3 EVALUATION SETTINGS

For all methods, we randomly select $N = 100$ queries for evaluation, and generate $K = 10$ responses per query. Our experiments are conducted using models from the `GPT-5*` family OpenAI (2025), one of the most advanced and influential LLMs to date. We evaluate two model configurations with reasoning capabilities, `GPT-5-mini`, and `GPT-5.1` to reflect realistic deployment scenarios under varying budget constraints. We set the temperature to 1, consistent with the default and non-modifiable decoding setting. For semantic similarity, we use the OpenAI `text-embedding-3-small`. Other settings are provided in the Appendix D.3.

## 6.4 MAIN RESULTS

Our main results are shown in Table 1. The plots illustrating the diversity–quality trade-off are presented in Fig. 1(Right), with additional results provided in Appendix E. Our main findings are summarized as follows:

- **Simple RAG does not yield more diverse outputs**: Compared to *Independent Sampling LLMs*, it even exhibits a decrease in viewpoint diversity. This remains the case even when more diverse contexts are introduced, such as by shuffling context, diversity-aware re-ranking, or using multiple query rewrites (§ 6.2). These simple retrieval-based baselines fail to increase diversity and therefore do not improve the overall *Unified Score*.
- **Prompt-based strategies can increase diversity but often lead to notable quality degradation**: *List Generation* and *Verbalized Sampling* improve diversity at the cost of a substantial drop in quality, with the diversity–quality trade-off being more severe on weaker models and less pronounced on stronger models. *Iterative Generation* shows a smaller quality decline, but yields only limited gains in diversity. As a result, these methods yield only limited improvements in the overall *Unified Score*.
- **DIVERGE achieves the best trade-off performance**: it obtains the highest *Unified Score* across *all* backbone models and *all* dimensions. Specifically, compared to *Independent Sampling*, DIVERGE improves semantic diversity by **2.1×–2.7×** and viewpoint diversity by around **1.6×**, while maintaining comparable output quality, with only a marginal decrease ($\sim 0.04$) in the quality score, successfully improving diversity while maintaining high-quality outputs.

## 6.5 FURTHER ANALYSIS

**Ablation Study.** We analyze the effects of removing key components of DIVERGE, including (a) search grounding and (b) result refinement. As shown in Figure 6, removing either component leads to a noticeable degradation in *Quality*, which in turn results in a lower *Unified Score*. These results empirically demonstrate the effectiveness of both components in maintaining high-quality outputs and achieving a favorable diversity–quality trade-off.

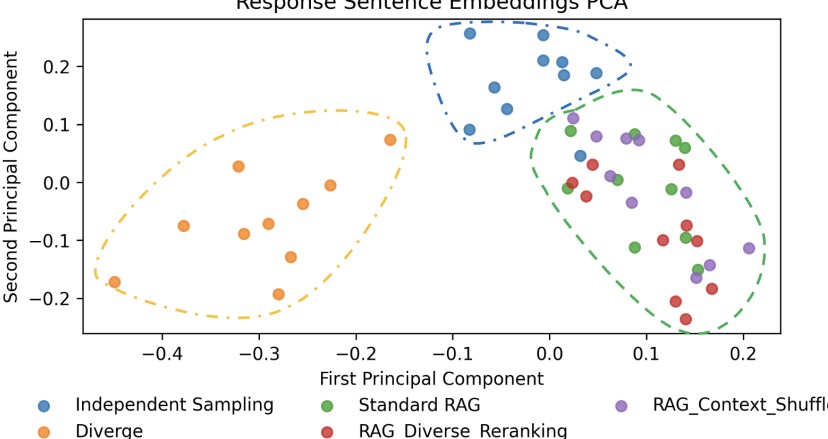

Figure 3: Responses to one query are projected into two dimensions using PCA over sentence embeddings. In this case, all responses are plausible. The visualization reveals three prominent clusters: homogeneous responses from direct LLM prompting; a separate but tightly grouped cluster from RAG and its variants, indicating that they differ from the LLM yet remain highly similar to each other; and a more diverse cluster corresponding to DIVERGE.

**Case Study.** As shown in Figure 3, responses to the query *"I have 10 years of experience in the web software development field. What can I do to improve my skills?"* clustered by Principal Component Analysis (PCA) to reduce sentence embeddings to two dimensions. We can clearly observe three distinct clusters: direct/independent prompting of the LLM forms a compact cluster (blue boundary) with highly similar responses; another cluster (green boundary) corresponds to RAG and its variants, indicating that while they differ from direct LLM outputs, they remain highly similar to each other; the final cluster (orange boundary) corresponds to DIVERGE, which exhibits substantially more diverse responses. This case study provides an intuitive illustration of the limitations of existing approaches and highlights the advantages of DIVERGE.

# 7 DISCUSSION

**Failure Analysis of Low-Quality Responses.** We analyze 30 low-quality cases produced by DIVERGE with respect to answer quality. We find that the primary issues fall into three categories: (1) diverting from the user's core intent, causing the response to miss key aspects of the question (40%); (2) overly generic recommendations that lack actionable specificity (30%); and (3) overly narrow focus on less important or peripheral aspects of the problem (17%). Detailed examples are provided in Table 3 in the appendix. These errors typically arise during the reflection over diverse viewpoints, where the selected viewpoints are either too broad, too narrow, or insufficiently relevant to the user's primary intent. As a result, while the generated responses may still offer some reference value, they fail to satisfy the user's information needs directly. This analysis highlights an important direction for future work: developing finer-grained control over viewpoint selection and integration during diversity-aware generation.

# 8 CONCLUSION

Our study highlights an important yet largely overlooked issue in typical RAG systems. Despite access to rich external knowledge, LLMs often fail to leverage different evidence from retrieved contexts, leading to highly homogenized behavior in RAG for open-ended information-seeking tasks. To address this concern, we propose DIVERGE, a plug-and-play agentic RAG framework specifically designed to ground retrieval and generation in diverse viewpoints for open-ended questions. Evaluated on real-world open-ended scenarios, DIVERGE substantially improves output diversity while preserving answer quality, enabling users to access a broader range of perspectives that can foster creativity and reduce the risk of overlooking underrepresented viewpoints. Our work paves the way for future research on open-ended question answering, encouraging consideration of diversity alongside answer quality.

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

# A APPENDIX

# B ADDITIONAL INFORMATION OF DIVERGE

## B.1 ALGORITHM

---

**Algorithm 1** DivRAG

---

**Require:** Generator $LM$, Retriever $\mathcal{R}$, Web Agent $\mathcal{W}$, Memory $\mathcal{M}$ Query $q$, Generation size $K$
    Response set $\mathcal{A} = \{a_1, \ldots, a_K\}$
1: Initialize $\mathcal{M} \leftarrow \emptyset$, $\mathcal{A} \leftarrow \emptyset$, $t \leftarrow 0$
2: Retrieved documents $d \leftarrow \emptyset$, Retrieval Index $\mathcal{I} \leftarrow \emptyset$
3: **while** $tK$ **do**
4:    **if** $t = 0$ **then**
5:      $\mathcal{I} \leftarrow \mathcal{W}(q)$
6:      $d \leftarrow \mathcal{R}(\mathcal{I}_0)$
7:      $a_0 \leftarrow LM(q, d)$
8:      $LM$ extracts initial viewpoints $v_0$ from $a_0$
9:    **else**
10:      $LM$ generates new viewpoint $v_t$ from memory $\mathcal{M}$
11:      $LM$ generates a viewpoint-conditioned query $q_t$
12:      $\mathcal{I} \leftarrow \mathcal{W}(q_t)$
13:      $d \leftarrow \mathcal{R}(\mathcal{I}, \mathcal{M})$
14:      $a_t \leftarrow LM(q, v_t, d)$
15:    **end if**
16:    $\mathcal{M} \leftarrow \mathcal{M} \cup \{(q_t, d_t, v_t, a_t)\}$
17:    $\mathcal{A} \leftarrow \mathcal{A} \cup \{a_t\}$
18:    $t \leftarrow t + 1$
19: **end while**

---

## B.2 PROMPT

---

**Summary prompt**

You are given a question and multiple existing answers.
Question: QUESTION
Existing answers: ANSWERS
Task: Identify the DISTINCT underlying views already present across the answers.
Guidelines: - A "view" refers to a perspective, framing, or stance — not wording. - Group answers that express the same core idea into one view. - If two answers differ only in phrasing, treat them as the same view. - Do NOT invent or infer new views.
Output requirements: - Output a LIST of views. - Each view must be a STRUCTURED ITEM with: - label: 2-5 words - description: exactly ONE sentence - Keep the list concise and non-redundant.
Output format (strict): Return ONLY a valid JSON array. Do NOT include explanations, comments, or markdown. [ "label": "...", "description": "..." , "label": "...", "description": "..." ]

---

**Reflection Viewpoint Prompt**

You are given an open-ended question and a list of views that have already been identified.
Question: QUESTION
Existing views: VIEWS
Task: Reflect on the coverage of the existing views and identify ONE new, meaningful direction that explores the original question from a new angle, while preserving its core constraints.
Guidelines: - The new view must remain relevant to answering the original question. - The new view should introduce a genuinely different angle without altering the question's intent or constraints. - The new view must be conceptually distinct from the existing views. - The new view should focus on an informative and helpful aspect of the question, rather than being overly generic or overemphasizing a minor detail. - Do NOT generate a full answer.
Output requirements: - Output exactly ONE new view. - Be concise and precise.
New view format (STRICT): "label": "...", # 2–5 words summarizing the new angle "description": "..." # exactly ONE sentence explaining how this angle helps address the question

**Query Generation Prompt**

You are generating a question that could reasonably be answered by the given answer.
Answer: ANSWER
Output MUST be valid JSON in the following format:
"question": "single concise question"
Rules: - Generate exactly one question. - Do NOT include explanations or multiple questions. - Do NOT add any text outside the JSON object. """

**Refine Prompt with View**

You are refining an existing answer to an open-ended question from a specific perspective.
Question: QUESTION
Perspective to prioritize: VIEW
Original answer: ANSWER
You are refining an existing answer to an open-ended question from a specific perspective, ensuring that the refined answer fully satisfies the original query and strictly follows all its instructions.
Specifically, the refined answer must: - Correct any statements that could be factually inaccurate or misleading - Ensure that claims are reasonably explained or appropriately qualified, rather than asserted without support - Be internally consistent and logically coherent - Address the original Question directly, grounding the answer in the given perspective - You MAY use the given perspective as an entry point or framing device, but the answer must clearly connect back to and help resolve the original Question rather than remaining at the level of the perspective alone. - Strictly follow any explicit instructions in the original Question (e.g., listing items or giving examples); required elements must appear first, with any additional explanation afterward
Constraints: - Do NOT introduce new factual claims beyond what is already implied by the original answer - Do NOT shift the focus to topics that are not relevant to the original Question - Keep the answer concise, focused, and well-structured
Output: Provide ONLY the refined answer text.

```
Refine Prompt without View

You are refining an existing answer to an open-ended question.
Question: QUESTION
Original answer: ANSWER
Your task is to produce a refined answer that:
- Improves factual accuracy and avoids potential errors - Avoids strong claims unless they are
well-supported or clearly qualified - Is internally consistent and logically coherent - Remains
clearly relevant to the original Question
Instructions: - Do NOT introduce new factual claims that are not implied by the original
answer. - Keep the answer concise, focused, and well-structured. - Directly answer the
Question; do not repeat or rephrase it.
Output: Provide ONLY the refined answer text.
```

### B.3 OVERVIEW OF DIVERGE

DivRAG is an iterative retrieval-augmented generation framework designed to produce *diverse yet relevant* answers to open-ended questions by explicitly modeling historical retrievals and generated viewpoints.

**Initialization.** Given an input query $q$, DivRAG initializes a diversity memory that stores: (i) previously issued queries, (ii) generated answers, (iii) extracted viewpoints, and (iv) embeddings of retrieved documents. The embedding model, chunking strategy, and large language model (LLM) are configured globally and shared across iterations.

**First Iteration** ($t = 0$)**.** DivRAG begins with a standard retrieval-augmented generation step:

1. **Retrieval.** The input query $q$ is used to perform a web search. Retrieved documents are chunked, embedded, and indexed into a vector store (cached per query for efficiency).

2. **Diversity-Aware Reranking.** Retrieved documents are reranked using a diversity-aware postprocessor. Since no retrieval history exists at $t = 0$, ranking is primarily driven by relevance.

3. **Generation.** The LLM generates an answer grounded in the retrieved documents using a standard RAG prompt.

4. **View Summarization.** The generated answer is summarized into a set of high-level viewpoints, which serve as semantic anchors for subsequent iterations.

**Subsequent Iterations** ($t > 0$)**.** For each subsequent iteration, DivRAG explicitly encourages novel perspectives:

1. **View Generation.** A new viewpoint is generated by prompting the LLM with the original question and the set of previously explored viewpoints.

2. **Query Reformulation.** A new query is synthesized conditioned on the newly generated viewpoint, steering retrieval toward under-explored semantic regions.

3. **History-Aware Retrieval.** Documents are retrieved and reranked using the DivReranker, which balances: (i) relevance to the current query, (ii) diversity among documents selected within the current iteration, and (iii) dissimilarity to documents retrieved in earlier iterations.

4. **View-Conditioned Generation.** The LLM generates an answer grounded in the retrieved documents and explicitly framed from the specified viewpoint.

5. **Memory Update.** The new query, retrieved document embeddings, generated answer, and viewpoint are stored in memory.

**Termination.** The process repeats until a predefined number of generations $K$ is reached. DivRAG outputs a set of answers that are grounded in external evidence, diverse across semantic viewpoints, and non-redundant with respect to past retrievals.

### B.4 OVERVIEW OF SEARCH IN DIVERGE

We implement a lightweight and reproducible web search and document extraction pipeline to support retrieval-augmented generation.

**Query Processing.** Given a textual query, the system retrieves web pages using a DuckDuckGo-based search interface executed via a subprocess. For each query, the search module requests up to $2N$ candidate URLs to account for filtering and extraction failures, where $N$ is the target number of retained documents.

**Domain and Format Filtering.** To improve content quality and reduce noise, retrieved URLs are filtered by: (i) excluding social media and multimedia platforms (e.g., Twitter, YouTube, Instagram), (ii) removing PDF documents, and (iii) ignoring domains matching a predefined blocklist. Only standard HTML pages from non-blacklisted domains are processed further.

**HTML Content Extraction.** For each retained URL, the system downloads the corresponding web page and extracts raw textual content using an HTML parser. Script, style, and non-textual elements are removed prior to extraction. The remaining visible text is normalized by line stripping and concatenation.

Pages that fail to download, return access errors (e.g., HTTP 403), or yield insufficient content are discarded.

**Length Filtering.** Extracted documents are required to exceed a minimum character threshold to ensure sufficient informational content. Only documents satisfying this constraint are retained as retrieval candidates.

**Rate Control and Robustness.** To reduce the risk of request throttling and blocking, the pipeline enforces randomized delays between requests and executes all search operations in a subprocess-safe manner. Errors during search or extraction are logged and handled gracefully without interrupting batch processing.

**Batch Processing and Output.** For large-scale experiments, queries can be processed in batch from an input file. For each query, the system outputs a list of retrieved documents, including the source URL, extracted text, and document length. All results are stored in a structured JSON format with timestamps to ensure reproducibility and traceability.

## C ADDITIONAL INFORMATION ON METRIC

### C.1 DETAILS OF VIEWPOINT DIVERSITY

**Embedding-Based Unique Claim Counting.** Given a set of generated texts and their corresponding embedding vectors, we estimate the number of semantically unique claims using a greedy pairwise similarity filtering procedure.

The algorithm iterates through the texts sequentially. For each text, its embedding is compared against the embeddings of all previously selected unique texts using cosine similarity. If the similarity with any existing unique embedding exceeds a predefined threshold $\tau$, the text is considered semantically redundant and discarded. Otherwise, it is added to the set of unique claims.

Formally, a text $x_i$ with embedding $\mathbf{e}_i$ is retained if

$$\max_{j \in \mathcal{U}} \cos(\mathbf{e}_i, \mathbf{e}_j)\tau,$$

where $\mathcal{U}$ denotes the index set of previously accepted unique texts. The final number of unique claims is defined as the size of $\mathcal{U}$.

This greedy pairwise filtering approach ensures that all retained claims are mutually dissimilar beyond the similarity threshold, providing an embedding-level approximation of semantic diversity.

---

**Claim Extraction Prompt**

You are an information extraction assistant.
Your task is to decompose an answer into a small set of high-level claims. Each claim must represent a complete, self-contained answer to the original question.
Question: QUESTION
Answer: ANSWER
Definition of a claim: - A claim must be able to stand alone as a reasonable answer to the question. - Each claim should express a complete position, recommendation, or conclusion. - A claim may summarize multiple supporting reasons, but should not list them separately. - Claims should be distinct alternative answers, not sub-points or justifications.
Guidelines: - Extract only claims that directly answer the question. - Do NOT extract supporting arguments, evidence, examples, or implementation details as separate claims. - Do NOT split a single answer into multiple claims if they jointly express one position. - If multiple sentences together express one answer, merge them into one claim. - Prefer fewer, higher-level claims over many fine-grained ones.
Output MUST be valid JSON in the following format:
"claims": [ "Complete answer-level claim 1", "Complete answer-level claim 2", "...", "Complete answer-level claim N" ]
Rules: - Each claim must be a single complete sentence. - Each claim must independently answer the question. - Each claim should be very concise. - Do NOT include numbering, labels, or text outside the JSON object.

---

## C.2   DETAILS OF QUALITY SCORE

---

**Quality LLM-As-A-judge Prompt**

You are evaluating an answer to an open-ended question. There is no single correct answer; instead, many different answers can be valid. An answer should be considered good if it is helpful or informative for some readers.
Question: QUESTION
Answer: ANSWER
Your task is to assess the quality of the answer along the following dimensions: 1. Factual accuracy: Does the answer contain factual errors? 2. Evidence support: Are the claims in the answer reasonably explained, rather than asserted without justification? 3. Internal consistency: Is the answer logically consistent with itself? 4. Question relevance: Does the answer provide information or insights that are helpful for addressing the question?
Based on these dimensions, assign ONE of the following verdicts: - Excellent: Fully addresses the question; accurate, well-supported, and internally consistent. - Good: Addresses the question well; mostly accurate with only minor issues. - Fair: Addresses the core of the question but has noticeable factual, support, or clarity issues. - Poor: Attempts to address the question but is largely incorrect, weakly supported, or unclear. - Irrelevant: The response does not address the question and provides no useful information.
Output MUST be valid JSON in the following format:
"verdict": "Excellent — Good — Fair — Poor — Irrelevant", "reason": "one short sentence or NONE"
Rules: - Choose exactly one verdict. - Focusing on some aspects or perspectives should not be treated as a weakness if it is relevant and helpful to the question. - If the answer does NOT address the question, verdict MUST be "Irrelevant". - The reason field MUST describe the main weakness or deficiency of the answer. - Keep the reason concise (max 15 words). - If the verdict is "Excellent", set reason to "NONE". - Do NOT output anything outside the JSON object.

---

## C.3   DETAILS OF QUALITY MODEL-HUMAN AGREEMENT

See Figure 4 for more information on the distribution.

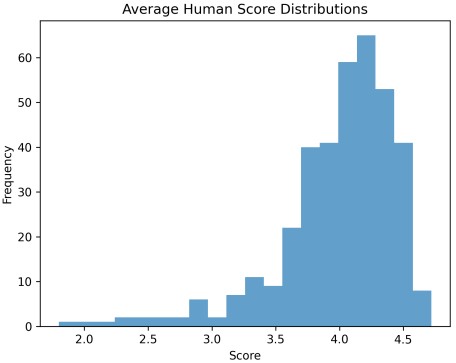

Figure 4: Average Human Score Distribution

# D  ADDITIONAL INFORMATION ON EXPERIMENT SETUP

## D.1  DETAILS AND EXAMPLES OF DATASET

We construct a curated subset of open-ended conversational prompts from the `Infinite-Chats-Taxonomy` dataset to support controlled diversity experiments.

**Source Dataset.** We start from the training split of the `liweijiang/infinite-chats-taxonomy` dataset. Each data instance consists of a multi-turn conversation and a set of annotated task categories.

**Prompt Extraction.** For each conversation, we extract the user prompt by selecting the first message whose role is labeled as `user`. All other conversational context is discarded. This results in a single prompt string per instance.

**Category Processing.** Each instance is associated with a list of category annotations. We extract the category labels from the annotation metadata and store them as a flat category list for each prompt.

**Predefined Category Filtering.** To focus on open-ended and opinion-diverse tasks, we define a predefined set of ten high-level categories: *Problem Solving, Decision Support, Concept Explanations, Skill Development, Recommendations, Opinion-Based Questions, Value-Laden Questions, Controversial Questions, Ideation and Brainstorming,* and *Personal Advice.*

An instance is retained only if *all* of its annotated categories belong to this predefined set. Formally, let $\mathcal{C}_i$ denote the category list of instance $i$ and $\mathcal{P}$ the predefined category set. Instance $i$ is selected if:

$$\mathcal{C}_i \subseteq \mathcal{P}.$$

**Subset Construction.** We iterate through the dataset sequentially and collect instances satisfying the category constraint until reaching a fixed budget of 200 examples. The resulting subset is stored as a standalone dataset for downstream experiments.

**Persistence.** The filtered dataset is serialized to disk using the HuggingFace `DatasetDict` format to ensure reproducibility and efficient reuse.

For examples in the dataset, please refer to Table 2

| Prompt | Categories |
|---|---|
| TrueNAS: Is there any benefit to creating a separate pool for data that is irreplaceable, or is it better to just add an additional backup for that dataset? | Problem Solving; Decision Support; Concept Explanations |
| I have 10 years of experience in web software development field. What can I do to improve my skill? | Skill Development; Personal Advice; Recommendations |
| What's the best way to switch scenes behind a closed elevator door in Blender without using a video editor? | Problem Solving; Skill Development |
| Here's some Lua code of a Factorio mod. Can you find some mistakes and fix it? | Problem Solving; Skill Development; Decision Support |
| What is the best way to do day trading from 100 dollars? | Skill Development; Problem Solving; Decision Support; Recommendations |
| What is the best business to do with 1000 Canadian dollars? | Decision Support; Recommendations; Problem Solving |
| What is a meal with good macros from Taco Bell that does not contain beans? | Recommendations; Problem Solving; Decision Support |
| Can you make a tax calculator only using methods or features specific to Ruby? | Problem Solving; Skill Development; Ideation and Brainstorming |
| I want to be better at using my Behringer RD-9 Analog Drum Machine as an instrument. Please write me a plan. | Skill Development; Problem Solving |
| Best programming language for open source contribution. | Opinion-Based Questions; Recommendations; Ideation and Brainstorming |
| Please explain entropy in simple terms that even a 14-year-old can understand. | Concept Explanations; Skill Development |
| If I want to avoid porn websites, which specific website should I avoid the most? | Personal Advice; Recommendations; Value-Laden Questions |
| Give me the key point of the book "The 5 Second Rule". | Concept Explanations |
| What are some of the cheapest mountains to climb in the world? | Recommendations; Decision Support |
| Why can't an image linked in an HTML file be read in Tomcat when a filter is enabled? | Problem Solving; Concept Explanations |
| What are the most fun things to do in Southampton? | Recommendations; Opinion-Based Questions |
| What websites sell alternative clothes in the UK suitable for a 32-year-old man wanting to dress more adventurously? | Recommendations; Personal Advice |
| Find at least five methodologies for regression, classification, and unsupervised learning tasks. | Problem Solving; Skill Development; Concept Explanations |
| Make a program that gives you bitcoin to your Coinbase wallet. | Problem Solving; Skill Development |
| Recommend free learning materials for beginners in reverse engineering. | Recommendations; Skill Development |

Table 2: Example prompts from the dataset and their associated categories.

## D.2 BASELINES

---

### Baseline LLM Prompt

You are a response generation assistant for open-ended questions. There is no single correct answer. Your goal is to generate multiple diverse, reasonable answers to the same question.
Question: QUESTION
Output MUST be valid JSON in the following format:
"answers": [ "Answer 1", "Answer 2", "...", "Answer K" ]
Rules: - You MUST produce EXACTLY K answers — no more, no fewer. - Each array element must be a single complete answer. - Ensure the output is valid JSON. - Do not use any quotation marks (") that appear inside answers. - Do NOT include numbering, bullet points, or labels inside the answers. - Do NOT output anything outside the JSON object.

---

### Verbalized Sampling baseline Prompt

You are a response generation assistant for open-ended questions. There is no single correct answer. Your goal is to generate multiple diverse, reasonable answers to the same question. Each response must be sampled at random from the full output distribution, rather than selecting the most likely or safest answers.
Question: QUESTION
Output MUST be valid JSON in the following format:
"answers": [ "text": "Answer 1", "probability": Probability 1 , "text": "Answer 2", "probability": Probability 2 , ... "text": "Answer K", "probability": Probability K ]
Rules: - You MUST produce EXACTLY K answers — no more, no fewer. - Each answer must be a single complete response to the question. - Each probability must be a numeric value between 0 and 1. - Probabilities do not need to sum to 1. - Ensure the output is valid JSON. - Do NOT include quotation marks (") inside the text fields. - Do NOT include numbering, bullet points, or labels inside the text. - Do NOT output anything outside the JSON object.

---

### RAG Multi Query Expansion baseline Prompt

You are a query expansion assistant for information retrieval.
Your task is to rewrite the original query into multiple distinct queries that can be used to retrieve complementary and diverse information.
The original query: QUERY
Output MUST be valid JSON in the following format:
"queries": [ "Expanded query 1", "Expanded query 2", "...", "Expanded query K" ]
Rules: - You MUST produce EXACTLY k queries — no more, no fewer. - Do NOT include numbering, bullet points, or labels inside the queries. - Do NOT output anything outside the JSON object. """

---

## D.3 DETAILS OF SETUP

We conducted our experiments using APIs obtained via `https://platform.openai.com/`. Detailed information about the APIs can be found on the website. The total API cost for all experiments was approximately $570.

For retrieval, we set the final Top-$K$ to 5. For web-based search, we retrieve between 5 and 10 documents per query, continuing the search until a sufficient number of valid documents is collected. All web data were collected in January 2026.

We apply a minimum document length threshold of 128 characters, and documents shorter than this threshold are filtered out. For diversity-aware retrieval, we initially retrieve 20 documents and apply reranking thereafter. In the reranking stage, we set the relevance–diversity trade-off parameters to $\alpha = 0.7$ and $\beta = 0.2$.

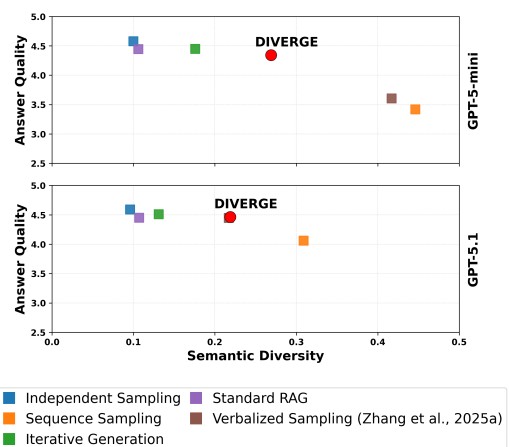

Figure 5: Semantic diversity–quality trade-off of different methods. Upper-Right indicates better.

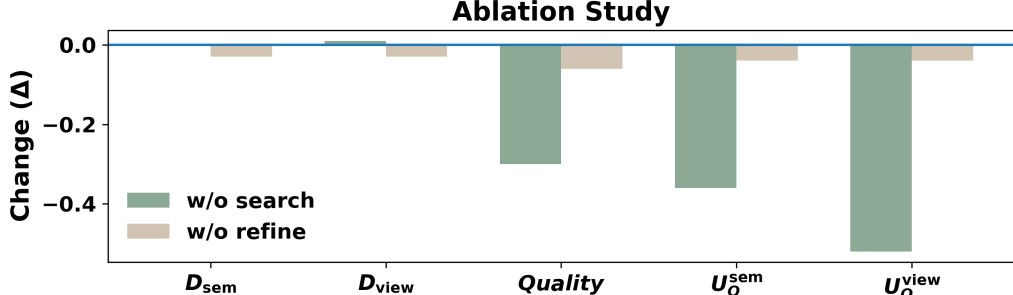

Figure 6: Ablation study of DIVERGE showing the impact of removing search grounding and result refinement on the performance of *Diversity*, *Quality*, and *Unified Score* on the *GPT-5-mini* model, highlighting the contributions of these components in DIVERGE.

For document chunking, we use a chunk size of 512 tokens with an overlap of 50 tokens. For viewpoint diversity evaluation, we set the similarity threshold $\tau$ to 0.75.

# E    SUPPLEMENT EXPERIMENTAL RESULTS

**Corelation Analysis.** We analyze the query-level correlations among semantic diversity, viewpoint diversity, and answer quality (Figure 7). Overall, we observe a negative correlation between diversity and quality, while the two diversity metrics are positively correlated, consistent with our expectations. We further examine cases where the two diversity metrics disagree and find that cases where viewpoint diversity much exceeds semantic diversity typically contain more claims (Figure 8 & Section 5). These patterns support our hypothesis that viewpoint diversity is more sensitive to, and thus better captures, intra-response diversity, such as when there are multiple claims inside the response.

Additional trade-off figure is shown in Figure 5. Another trade-off figure is shown in Figure 1. While some models exhibit higher diversity in this figure, their generation quality degrades substantially. According to the results in Table 1, when compared using the *Unified Score*, our method still achieves superior overall performance.

# F    FAILURE ANALYSIS

Please refer to Table 3 for details.

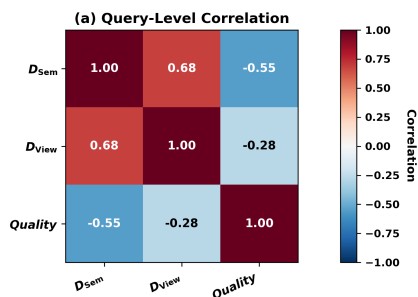

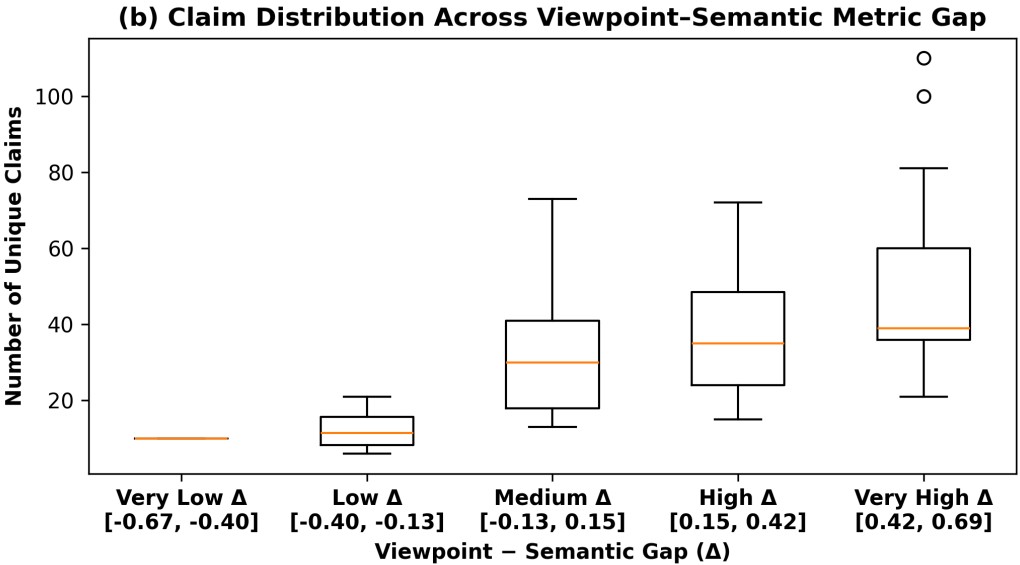

Figure 8: Responses with higher viewpoint (vs. semantic) diversity contain more distinct claims, while higher semantic (vs. viewpoint) diversity corresponds to fewer claims.

| Error Type | Question | Answer | Explaination |
|---|---|---|---|
| Diverting from the user's core intent (40%) | What's the best way to switch scenes behind a closed elevator door in blender without using a video editor | Use Blender's own Video Sequence Editor (VSE) and do the switch there, timed exactly to the frames where the elevator doors are fully closed, so everything stays inside the same .blend and you don't need an external editor. 1. **Prepare the two parts of the animation** - Part A: before the elevator ride / before the change. - Part B: after the elevator ride / in the new environment. You can: - Render each part, then use those renders in the VSE, or - Add them directly as **Scene** strips in the VSE (no external files needed). 2. **Open the Video Sequence Editor** - Switch to the **Video Editing** workspace (or open a VSE editor area).... | Failure to follow not using a video editor |
| Diverting from the user's core intent (40%) | What websites sell alternative clothes in the UK that would suit a 32-year-old man who mostly wears checked shirts with jeans and wants to dress more adventurously? | If you want to dress more adventurously but aren't sure what to buy, UK style-box and rental services are a good way to test things before committing... | Shifts away from alternative clothes to style-box and rental services |
| Too general (30%) | What websites sell alternative clothes in the UK that would suit a 32-year-old man who mostly wears checked shirts with jeans and wants to dress more adventurously? | - **r/malefashionUK (Reddit)** - **r/malefashionadvice (Reddit)** – global, but with plenty of UK posters. - Search tags and threads mentioning UK retailers and "edgier" or "alternative" looks; you'll find links to online shops that ship easily to the UK. - When you find pieces you like, you can post "fit checks" to see if they look age-appropriate for a 32-year-old. - **Instagram hashtags and tags** – use them as a visual way to find both styles and shops. - Try hashtags like: - '#ukmensstyle', '#britishmenswear' - '#alternativefashionmen', '#indiefashionmen' - '#streetwearuk', '#darkwear' | These Suggestions are too general, should focus on alternative clothes |
| Too narrow and over-focused on unimportant details (17%) | What are the security concerns when travelling to Italy? | When travelling to Italy, important security concerns from a natural and environmental standpoint include: 1. **Heatwaves (especially in summer, particularly in the south and major cities)** - High summer temperatures can raise the risk of heat exhaustion and heatstroke. - Security concern: dehydration, fainting, and other heat-related illnesses can become medical emergencies, especially for children, older adults, and people with health conditions. | Too narrow focus on Heatwaves, which is not very important on security concerns |

Table 3: Illustrative examples of typical errors of DIVERGE in open-ended question answering.

## G CLAIMS EXAMPLE

Please see Table 4 for more information about examples of responses and claims.

## H LIMITATION

On the one hand, evaluating responses to open-ended questions is inherently challenging, particularly for knowledge-seeking tasks, which we argue are difficult even for human experts. As a result, LLM-as-a-judge evaluations may not always provide perfectly accurate quality assessments, leaving room for improvement in quality evaluation methods. On the other hand, it remains unclear which

| Query | Response | Claims |
|---|---|---|
| I have 10 years of experience in the web software development field. What can I do to improve my skills? | With a decade of experience, shift from "getting things done" to deliberate, measurable skill growth: deepen fundamentals, broaden selectively, and build sustainable habits. Practical, prioritized plan: 1. Set concrete goals - Pick 2–3 outcomes (e.g., become an architecture/tech lead, master frontend performance with React, or own cloud-native production systems). - Define measurable milestones (projects, certifications, internal KPIs, or interview targets). 2. Deepen fundamentals and code quality - Regularly read and refactor code: reduce duplication, improve abstractions, apply appropriate design patterns. - Practice writing well-tested code (TDD where useful; unit/integration/e2e tests) and record design decisions (notes or ADRs). 3. Learn production systems and architecture - Study system design, scalability, observability, performance tuning, security basics, and fault tolerance. - Strengthen cloud and infra skills: containers, orchestration (e.g., Kubernetes), at least one cloud provider, CI/CD, and monitoring practices. 4. Broaden your stack selectively - Add one complementary stack or language on a 6–12 month cadence (e.g., backend if you're frontend-heavy, or modern frontend if you're backend-heavy). - Focus on when and why to choose tools, not only how to use them. 5. Read and study expert code; contribute to OSS - Read well-maintained open-source repositories to absorb idioms and architecture. - Start with small PRs, then work toward larger contributions or maintenance. 6. Practice problem solving lightly and regularly - Do short, regular algorithmic exercises to keep reasoning sharp—useful for debugging complex issues and interviews. 7. Teach, mentor and get feedback - Mentor juniors, run code reviews, and pair program to surface blind spots. - Teach publicly (blog posts, talks, workshops) to clarify thinking and identify gaps. 8. Build end-to-end projects - Own projects that require architecture, deployment, CI, monitoring, and scaling. Instrument them and measure performance and errors. 9. Keep curated learning habits - Daily/weekly microlearning (30–60 min/day); a weekly or biweekly deeper session (several hours). - Follow curated sources and read a technical book or long-form article periodically. 10. Career and communication skills - Develop system-design, interviewing, leadership, and product-thinking skills if moving toward senior/architect/manager roles. - Practice explaining tradeoffs to non-technical stakeholders. 11. Rest, review and iterate - Schedule breaks to avoid burnout. Review progress regularly (monthly or quarterly) and adjust goals. Example cadence - Daily: 30–60 min coding/reading. - Weekly: coding katas + focused refactoring/OSS work. - Monthly: write or present a short post; reach a project milestone. - Quarterly: complete a larger project or reassess goals. Useful tools & resources (examples) - Code quality: linters, formatters, code scanners. - Dev tooling: quality editor, Docker, CI/CD tools, Git workflows. - Cloud: fundamentals of a major cloud provider and orchestration. - Learning: curated courses, books, and community venues (blogs, meetups). If you want, tell me your primary stack and career goal (tech lead, architect, IC specialist, or founder) and I'll draft a 3-, 6- and 12-month learning plan with specific resources and milestones. | 1. Set concrete goals 2. Deepen fundamentals and code quality 3. Learn production systems and architecture 4. Broaden your stack selectively ... 11. Rest, review and iterate |

Table 4: Examples of Responses and Claims

diversity metrics best align with human perception. We encourage future work to incorporate human feedback.

