# OpenReview forum: "DIVERGE: Diversity-Enhanced Retrieval-Augmented Generation for Open-Ended Information Seeking"
_ICLR.cc/2026/Workshop/AFAA — Submitted to AFAA 2026_

### Official Review · Reviewer_cXxz · 2026-02-16
**DIVERGE: Diversity-Enhanced Retrieval-Augmented Generation for Open-Ended Information Seeking**

**Rating:** 4
**Confidence:** 5

**Summary:**

This paper proposes DIVERGE (Diversity-Enhanced Retrieval-Augmented Generation), a plug-and-play agentic RAG framework designed to improve diversity in open-ended information-seeking tasks
. The authors identify that conventional RAG systems suffer from single-answer bias, where increased retrieval diversity does not translate into diverse outputs
To address this, DIVERGE introduces reflection-guided viewpoint generation and viewpoint-aware diversity retrieval to encourage exploration of multiple perspectives while reducing redundancy
. The framework also includes a lightweight memory mechanism to preserve diversity across iterations
The paper further proposes evaluation metrics to measure the diversity–quality trade-off
. Experiments on the Infinity-Chat benchmark demonstrate that DIVERGE improves semantic and viewpoint diversity while maintaining comparable answer quality

**Strengths:**

Strengths

Clearly identifies the single-answer bias problem in RAG systems
Novel reflection-guided viewpoint generation mechanism
Plug-and-play design compatible with existing RAG pipelines
Introduces dedicated metrics for diversity–quality evaluation
Strong empirical validation on Infinity-Chat benchmark

**Weaknesses:**

Focuses mainly on open-ended tasks; generalization to other domains is unclear.
Additional reflection and retrieval steps may increase computational cost.
Diversity gains may depend on prompt or task design.
Evaluation relies heavily on a single benchmark (Infinity-Chat).

---

### Official Review · Reviewer_SSLE · 2026-02-21
**DIVERGE A thoughtful agentic RAG approach for diversity in open ended information seeking**

**Rating:** 3
**Confidence:** 4

**Summary:**

This paper studies an important and under explored problem in modern retrieval augmented generation systems, namely the tendency of both LLMs and RAG pipelines to collapse toward a single dominant answer even when user queries are open ended and admit multiple plausible viewpoints. The authors carefully analyze why existing RAG approaches fail to translate retrieval diversity into generation diversity, and identify key limitations such as single answer bias and lack of long horizon diversity preservation.

To address this, the paper proposes DIVERGE, a plug and play agentic RAG framework that explicitly models viewpoints through an iterative process combining reflection guided viewpoint generation, viewpoint aware retrieval, and viewpoint conditioned generation. The framework is designed to work with closed source frontier models without relying on token level logits. In addition, the authors introduce new evaluation metrics for diversity quality trade offs in open ended information seeking, including semantic diversity, viewpoint diversity, and a unified harmonic score. Experiments on the Infinity Chat dataset demonstrate that DIVERGE achieves substantially higher diversity while largely preserving answer quality compared to strong baselines

**Strengths:**

The paper tackles a real and timely problem that clearly aligns with the goals of the AFAA workshop, especially around pluralism, fairness, and avoiding epistemic collapse in aligned and agentic systems. The motivation is strong and well grounded in both recent literature and practical deployment realities of RAG systems. The conceptual framing around viewpoints as a first class abstraction is intuitive and well explained, and the agentic design feels natural rather than forced.

Methodologically, the work is careful and systematic. The authors do not simply propose a new method but also analyze why existing approaches fail, which strengthens the overall contribution. The framework is practical in that it avoids reliance on decoding tricks or internal logits, making it relevant for real world systems using closed models. The evaluation is thorough for a workshop paper, with clear baselines, ablations, and qualitative analysis. The proposed diversity metrics, especially viewpoint diversity, are thoughtful and address known shortcomings of purely semantic measures. Overall the paper feels mature, technically solid, and well aligned with the workshop theme.

**Weaknesses:**

While the ideas are strong, the paper sometimes leans heavily on LLM as judge evaluations for quality, which can raise concerns about subjectivity and metric robustness, even though some human agreement analysis is provided. The Infinity Chat dataset is appropriate but still limited in scope, and it is not fully clear how the approach would behave in more sensitive fairness critical domains such as political or social decision making where diversity may interact with harm.

The agentic pipeline is conceptually clean but also somewhat complex, and it would help to better understand the computational or latency costs introduced by the iterative process. Some design choices such as viewpoint extraction and claim decomposition rely on LLM prompts that may themselves introduce bias or instability, and this is not deeply examined. Finally, while the paper argues for fairness and inclusivity, the connection to formal fairness definitions remains mostly conceptual rather than operational.

---

### Official Review · Reviewer_pc9n · 2026-02-23
**DIVERGE: Improving Diversity in Open-Ended RAG**

**Rating:** 4
**Confidence:** 3

**Summary:**

The paper identifies a limitation of standard RAG systems in open-ended information-seeking settings: despite diverse retrieval, generated outputs often collapse to a single dominant answer. The authors propose DIVERGE, a plug-and-play agentic RAG framework that combines reflection-guided generation and memory-augmented iterative refinement to promote viewpoint diversity while maintaining quality. They introduce new metrics for evaluating the diversity–quality trade-off and validate them against human judgments. Experiments on the Infinity-Chat dataset show improved diversity while preserving response quality compared to strong baselines.

Note: It looks like the authors are using the 2025 template of ICLR, in the paper this is mentioned: Under review as a conference paper at ICLR 2025

**Strengths:**

1. Clearly articulated and practically relevant problem.
2. Coherent and well-motivated framework for promoting diversity in RAG.
3. Introduces diversity–quality evaluation metrics with reported human correlation.
4. Strong empirical results on a real-world dataset.

**Weaknesses:**

1. Evaluation is limited to a single dataset (Infinity-Chat), leaving questions about generalizability across domains.
2. The proposed diversity–quality metrics are validated only within the presented experimental setting.
3. Experiments rely primarily on GPT-family models, and evaluation on additional model architectures would strengthen claims of general applicability.

---

### Official Review · Reviewer_6yhs · 2026-03-02
**A Diverse viewpoint seeking enhancement for RAG especially for open ended tasks**

**Rating:** 4
**Confidence:** 4

**Summary:**

The paper proposes a framework that enhances retrival of information especially focusing on diversity of information to present diverse and possibily more balanced perspectives. The paper presents homogeneity of LLMs as a major challenge that could lead to epistemic collapse and puts their Diverse framework as a method to promote increased generation diversity.

The Diverge method which is inspired by Wang et al's plan and solve based design applies a method of reflection guided by diverse viewpoint generation. Once a set of view points are generated, a relfection is done on these viewpoints iteratively to generate new less explored viewpoints. To overcome the limitation of an LLM having limited information about viewpoints, the paper suggests using web queries for allowing incorporation of multiple view points beyond the capability of the current model.

The paper's methods, evaluation and experimental evidence are good and the failure analysis is also helpful to understand the shortcomings as well as the future directions for the work.

**Strengths:**

The paper uses a combination of diversity and quality metrics along with the acknolwedgement that quality needs to be maintained while retrieving diverse view points.  They use LLM as a judge for a four dimensional quality scoring and also employ human annotation. The use of a unified diversity quality harmonic score also seems well thought out.

The paper supplies good experimental evidence in the form of their method tested on OpenAI's GPT 5 models and their ablation studies are also helpful in understanding that both phases of their method are required for effective diverse rag retrieval.

**Weaknesses:**

The paper presents this method as overcoming challenge C3 of overcoming practical applicability as decoding strategy based methods may not be exposed by APIs. This is reasonable however they should provide a comparison of their method to decoding strategy methods. While their method is in an entirely different league, it would provide an helpful comparison. Perhaps the method can be applied to an open source model as well with scores for quality and diversity for both Diverge and decoding strategy methods.

---

### Meta-Review · Area_Chair_okAo · 2026-02-24

**Recommendation:** Main Papers Track
**Confidence:** 4

**Metareview:**

There is an overall consensus on the timeliness of the paper and its proposed approach. The collapse to a single answer goes against pluralism and produces biased answers, which the paper tackles. While these positive points make the paper great fit to the workshop, a common weakness was highlighted which is the reliance on a single task for evaluation. The use of LLM-as-a-judge also seems to be a concern.
Overall, I think the paper is a good fit for the workshop and I am leaning towards acceptance.

P.S: Issues regarding template and github link  should be fixed for the camera ready.
In particular, the link does not work and was not fully anonymized, however since it does not contain authors names I think it is not a big issue.

---

### Decision · Program_Chairs · 2026-03-02

**Decision:**

Reject

**Comment:**

As brought to our attention, the paper breaks the double-blind requirements due to the presence of (a) github link to a repository, even though the repository is not available, it gives information about the authors' affiliations, and (b) the presence of author names in the file 'src/divrag.py' on the anonymous code release. Thus, despite strong contributions, the paper has been rejected from the workshop. We hope the authors will still benefit from the reviews.